# Mechanical Properties of L-Shaped Column Composed of RAC-Filled Steel Tubes under Eccentric Compression

**Tengfei Ma [1,2], Zhihua Chen [1,3], Yansheng Du [1,3,\*], Ting Zhou [4] and Yutong Zhang [1]**

[1]  School of Civil Engineering, Tianjin University, Tianjin 300072, China; mtf@nciae.edu.cn (T.M.); zhchen@tju.edu.cn (Z.C.); zhangyt@tju.edu.cn (Y.Z.)
[2]  Department of Civil Engineering, North China Institute of Aerospace Engineering, Langfang 065000, China
[3]  Key Laboratory of Coast Civil Structure Safety of Ministry of Education, Tianjin University, Tianjin 300072, China
[4]  School of Architecture, Tianjin University, Tianjin 300072, China; zhouting1126@126.com
\*  Correspondence: duys@tju.edu.cn

**Abstract:** In this paper, the eccentric compression test of seven specimens was conducted to explore the application possibility of recycled aggregate concrete (RAC) in L-shaped columns composed of concrete-filled steel tubes (CFST). The main parameter is the replacement ratio of recycled coarse aggregates (RCA), and an L-shaped column of composed hollow steel tubes was set as a control. The test results indicated that the bearing capacity and stiffness of the L-shaped column composed of RAC-filled steel tubes (RACFSTs) are better than those of the L-shaped column composed of hollow steel tubes. The compressive strength of concrete is reduced by 73.1% as the replacement ratio of RCA increases from 0 to 100%, while that of the column is merely reduced by 23.9%. The strength disadvantage of RAC is compensated by the confinement of steel tubes. Besides, the result of the eccentric compression test (80 mm eccentricity) was compared with the axial compression test (0 mm eccentricity). The increase in eccentricity reduced the bearing capacity and ductility due to additional bending moments. The finite element model (FEM) was established by software ANSYS to compare with the experimental results. The bearing capacity deviation of FEM is 4.23~6.56%. The parametric analysis was carried out to summarize the influence of parameters such as eccentricity, material strength, and steel tube thickness. With the increase of eccentricity, the bearing capacity of the RACFST decreases gradually. In engineering design, the bearing capacity of the RACFST can be improved by increasing the strength and thickness of the steel.

**Keywords:** L-shaped column; recycled aggregate concrete; concrete-filled steel tube; eccentric compression; finite element model





## 1. Introduction

With the development of urbanization, the construction space has put forward higher requirements for the traditional structure. In the traditional structure, the external corner of the frame column not only reduces the construction space but is also difficult to handle in terms of architectural aesthetics, see Figure 1a. The special-shaped columns are proposed to solve the above problems, which can completely hide the structural column in the infill wall. The special-shaped column avoids the negative impact of the external corner on the building function, as shown in Figure 1b. Due to the architectural advantages of special-shaped columns, researchers have investigated the mechanical properties of various special-shaped columns. Li et al. [1] carried out an axial compression test on the steel-reinforced concrete short column with a T-section, and the effect of shaped steel ratio and shear-span ratio on bearing capacity was explored. Tokgoz et al. [2] presented the eccentric compression test of L-shaped high strength reinforced concrete and concrete-encased composite columns, which indicated that the addition of steel fibers to high-strength concrete improved the mechanical properties of the columns under eccentric loading. Xue et al. [3–5] conducted

seismic damage tests on steel-reinforced concrete irregular section columns. The effects of different loading schemes, axial compression ratios, and steel distribution ratios on the seismic performance of specimens were revealed.

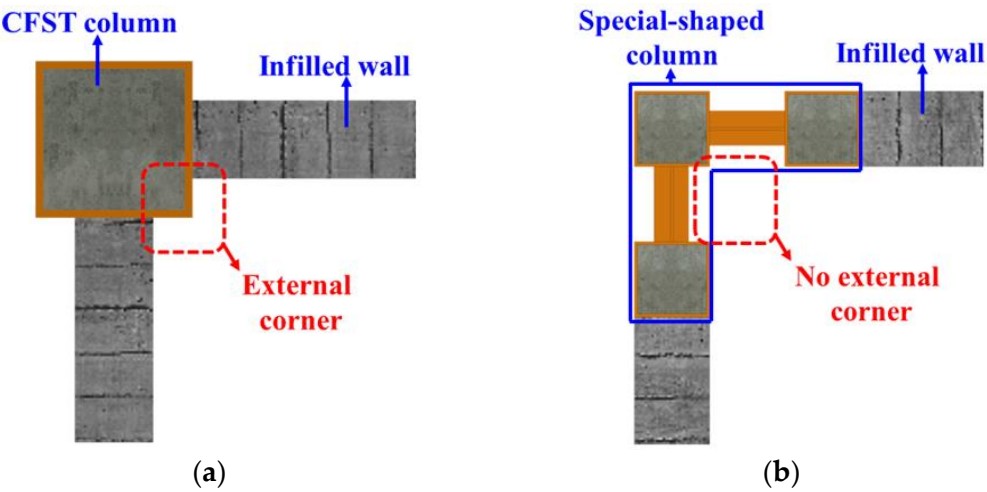

**Figure 1.** The comparison of the traditional column and the special-shaped column. (**a**) Traditional column. (**b**) Special-shaped column.

However, the steel-reinforced concrete columns require complex formwork in construction. The construction of concrete-filled steel tube (CFST) columns can avoid the use of complex formwork. The different types of special-shaped steel tubes are more convenient to prefabricate in the factory. Han [6–8], Tu [9,10], and Zhou [11,12] carried out mechanical tests of special-shaped CFST columns, respectively. The influence of different parameters on special-shaped CFST columns was explored in the tests. The appropriate design formula was also proposed. Li et al. [13] conducted a quasi-static test on the joint of T-shaped CFST columns and H-shaped steel beams to study the effects of the beam-column connection form and the axial compression ratio on its seismic performance. To further facilitate the prefabrication and construction of special-shaped mono CFST columns, special-shaped columns with small cross-sections were proposed by Chen et al. [14,15], as presented in Figure 2. The cross-section form of the column can be divided into L-shaped, T-shaped, and cross-shaped according to the structural requirements, corresponding to corner columns, side columns, and middle columns, respectively. This special-shaped column is formed by welding the mono CFST columns through the connection plates, which can ensure the collaborative work of the CFST columns. Rong et al. [15–18] conducted the axial compression tests of the special-shaped columns composed of the square CFST with different slenderness ratios. The test results summarized the strength failure and instability failure modes. The L-shaped column composited of the mono column connected with the perforated steel plate was proposed by Zhou et al. [19–22]. The axial compression test indicated that the L-shaped column connected by the perforated steel plate can still ensure cooperative deformation of the mono column, but the bearing capacity is reduced. By evaluating mechanical properties and processing complexity, the special-shaped columns connected with non-perforated steel plates are more suitable for the application. The previous research has demonstrated that the special-shaped column has high overall stiffness and strong energy dissipation capacity, which is very suitable for prefabricated high-rise buildings.

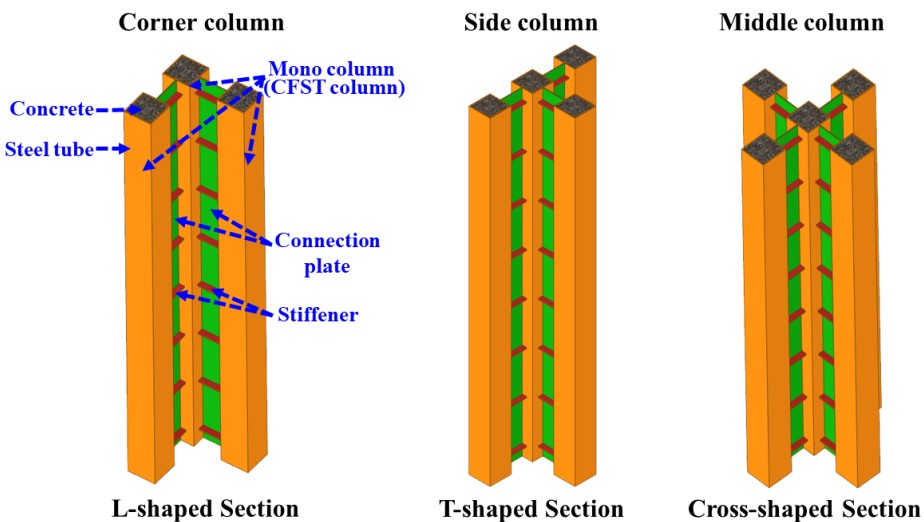

**Figure 2.** Special-shaped columns composed of mono columns.

Due to the continuous deterioration of the current global natural environment, the use of recyclable green building materials is an inevitable choice for the construction industry [23]. Recycled aggregate concrete (RAC) made from solid waste can not only reduce the mining of natural aggregates but also absorb construction waste. In China, construction waste accounts for 40% of the waste produced annually [24]. RAC has a great development space and application value. However, the lower strength and poorer elastic modulus of the RAC impeded the engineering application compared with ordinary concrete [25–27]. The reasons for the strength reduction of recycled concrete are: recycled aggregate has poor properties, the effect of the production process, the effect of the strength of parent concrete, and the effect of the repeated use of recycled aggregate [28]. To improve the mechanical properties of RAC, many scholars have studied the mechanical properties of RAC-filled steel tube (RACFST) columns by utilizing the confining effect of steel tubes [29]. In compression, the RAC undergoes lateral deformation due to Poisson's ratio effect. The steel tube produces passive confinement on the RAC, which can improve the compressive strength of the RAC. Meanwhile, the buckling of the steel tube is inhibited by the inner RAC. This is similar to ordinary CFST. In the application of recycled concrete-filled steel tubes, the initial defects caused by the strength, compactness, and pouring defects of recycled concrete should be reduced as much as possible. Wu et al. [30] conducted the quasi-static test of 15 thin-walled RACFST columns produced by large waste concrete blocks as recycled coarse aggregate (RCA). The test results presented that the bearing capacity of RACFST varies less when the replacement ratio of RCA varies between 0–40%. The bond behavior of CFST and RACFST were summarized by Lyu [31] based on the push-out test of 56 RACFST columns. The calculation formula was proposed for the bond strength between the concrete and steel tubes. Wang et al. [32] displayed that the reduction in compressive strength of RACFST is less than 10% compared to CFST for the same concrete mix ratio. The research on the mechanical properties of RACFST indicated that the confinement of steel tubes could compensate for the shortcomings of RAC. Therefore, it is feasible to apply RAC to the special-shaped column composed of mono CFST columns.

At present, the research on the innovative special-shaped columns composed of RACFST is still in shortage. To study the mechanical properties of shaped columns under the combined action of axial pressure and bending moments. In this paper, the eccentric compression test of the special-shaped columns composed of RACFST was conducted. Six specimens were designed with the different RCA replacement ratios of RAC as the parameter. A special-shaped column composed of hollow steel tubes was designed as a control specimen. The effects of the RCA replacement ratios of RAC on the mechanical properties of specimens were evaluated by comparing the failure modes and load-displacement curves of the tests. The strain analysis was applied to investigate the correspondence between the

overall deformation and the local strain of the columns. The finite element model (FEM) was compared with the experimental results. Parametric analysis was carried out based on the validated FEM. The effect of eccentricity, material strength, and steel tube wall thickness on the bearing capacity of the specimens was investigated.

## 2. Experimental Program

### 2.1. Specimens Design

The L-shaped columns composed of CFST were created in Cangzhou, China [33]. The dimensions of the specimens in this paper are 1/2 scale L-shaped columns used in actual engineering. The mono column is an RAC-filled steel tube. The height of the mono column was 1500 mm. The wall thickness of the steel tubes was 3.75 mm. The L-shaped column was composed of three mono columns connected together by two connection plates. To enhance the stiffness of the connection plate, stiffener ribs with an interval of 200 mm were welded on the connection plate. The dimensions of the stiffeners were 100 mm × 20 mm × 5.75 mm. The 20 mm thick cover plate was welded on both ends of the L-shaped column to prevent local damage at the column end during compression. The details are presented in Figure 3. The concrete used in the steel tube was RAC. Six specimens were fabricated with different RCA replacement ratios of RAC, 0%, 20%, 40%, 60%, 80%, and 100%. An L-shaped column composed of hollow steel tubes was set as the control specimens. All the parameters are shown in Table 1.

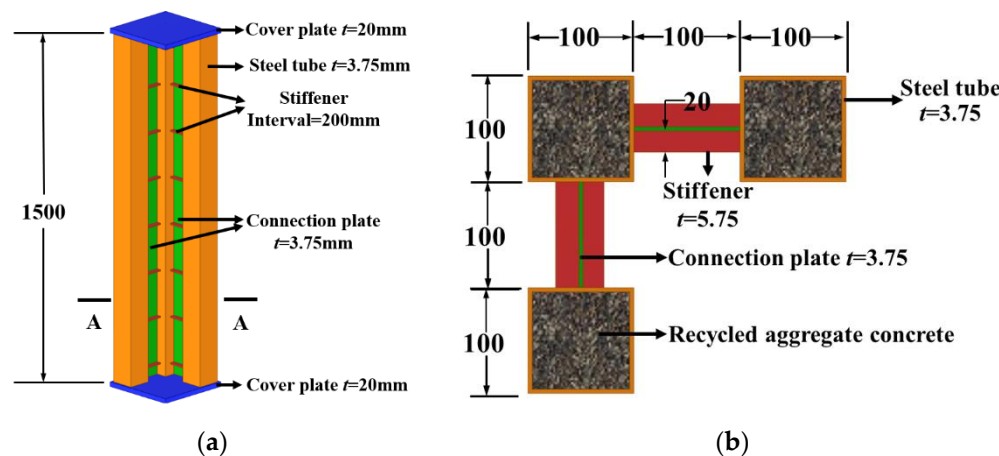

(**a**)          (**b**)

**Figure 3.** Dimensions of the L-shaped column (in mm). (**a**) General dimension. (**b**) Cross-section A-A.

**Table 1.** The detailed parameters of specimens.

| Specimens | Steel Tube $H \times l \times l \times t$ (mm) | Connection Plate $H \times l \times t$ (mm) | $f_{c,cube}$ (MPa) | Replacement Ratio of RAC | Eccentric Distance (mm) |
|---|---|---|---|---|---|
| P80-0 | 1500 × 100 × 100 × 3.75 | 1500 × 100 × 3.75 | 33.36 | 0% | 80 |
| P80-20 | 1500 × 100 × 100 × 3.75 | 1500 × 100 × 3.75 | 29.00 | 20% | 80 |
| P80-40 | 1500 × 100 × 100 × 3.75 | 1500 × 100 × 3.75 | 17.59 | 40% | 80 |
| P80-60 | 1500 × 100 × 100 × 3.75 | 1500 × 100 × 3.75 | 13.30 | 60% | 80 |
| P80-80 | 1500 × 100 × 100 × 3.75 | 1500 × 100 × 3.75 | 11.94 | 80% | 80 |
| P80-100 | 1500 × 100 × 100 × 3.75 | 1500 × 100 × 3.75 | 8.99 | 100% | 80 |
| P80-K | 1500 × 100 × 100 × 3.75 | 1500 × 100 × 3.75 | - | No RAC | 80 |

**Note:** $H$ is the height of the component, $l$ is the side length of the section, $t$ is the thickness of the steel plate, $f_{c,cube}$ is the cubic compressive strength of RAC.

### 2.2. Material Properties

2.2.1. Recycle Aggregate Concrete

The mix ratio of the ordinary concrete used in this paper was C30 grade according to specification JGJ55 [34]. The concrete was fine aggregate concrete. Medium sand with a particle size of 0.3 mm~0.5 mm was used. The fineness module was 2.6 mm. The particle size

of natural aggregate and recycled coarse aggregate was 5 mm~15 mm. The crushing index was 10%. The water absorption was 2.3%. Recycled aggregate concrete was prepared by replacing natural aggregates with different proportions of recycled coarse aggregates. The specific mix ratio is presented in Table 2. Three cubic blocks (100 mm × 100 mm × 100 mm) and three prism blocks (150 mm × 150 mm × 300 mm) for each RCA replacement ratio of RAC were tested to reduce the test error according to specification GB/T50081 [35]. The compressive strength and elastic modulus of RAC were obtained in Table 3. The standard compression test is presented in Figure 4.

**Table 2.** The detailed mix proportion of RAC (kg/m$^3$).

| Replacement Ratio | 0% | 20% | 40% | 60% | 80% | 100% |
|---|---|---|---|---|---|---|
| Water | 44.74 | 40.21 | 35.80 | 31.34 | 26.87 | 22.40 |
| Cement | 138.25 | 138.25 | 138.25 | 138.25 | 138.25 | 138.25 |
| Sand | 149.40 | 149.40 | 149.40 | 149.40 | 149.40 | 149.40 |
| Coarse aggregate | 267.60 | 214.08 | 160.56 | 107.04 | 53.52 | 0 |
| RCA | 0 | 58.00 | 115.98 | 173.96 | 231.95 | 289.94 |

**Table 3.** The material properties of RAC.

| Replacement Ratio | $f_{c,cube}$ (Mpa) | $f_{c,prism}$ (Mpa) | $E_c$ (Mpa) | Curing Age (Day) |
|---|---|---|---|---|
| 0% | 33.36 | 26.7 | 32,964 | 55 |
| 20% | 29.00 | 25.8 | 23,064 | 55 |
| 40% | 17.59 | 14.0 | 13,722 | 28 |
| 60% | 13.30 | 12.2 | 14,369 | 28 |
| 80% | 11.94 | 10.0 | 10,345 | 28 |
| 100% | 8.99 | 8.4 | 9912 | 28 |

**Note:** $f_{c,cube}$ is the cubic compressive strength of RAC; $f_{c,prism}$ is the Prismatic compressive strength of RAC; $E_c$ is the elasticity modulus of RAC. The concrete test block test was carried out on the same day as the specimen loading.

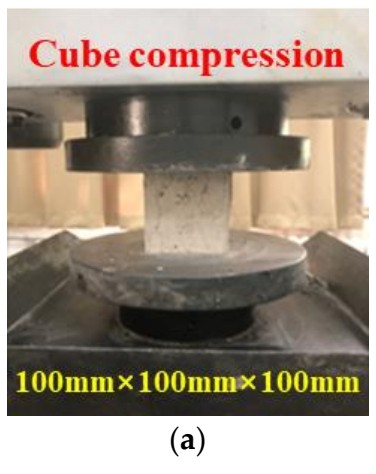

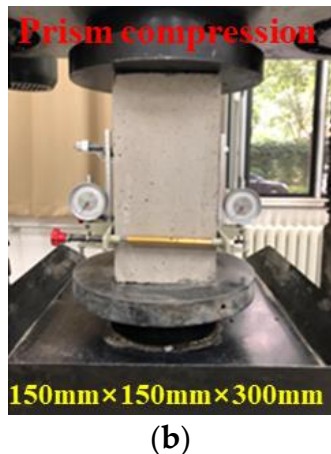

(**a**)　　　　　　　　　　　　　(**b**)

**Figure 4.** The compression test of the concrete. (**a**) Cube compression test. (**b**) Prism compression test.

### 2.2.2. Steel

According to specification GB/T228 [36], tensile tests were conducted for steel coupons taken from steel tubes, connection plates, and stiffener ribs. The dimension of the steel coupon and the test machine are illustrated in Figure 5. All the steel used in the specimens was Q235 grade. The mechanical properties of the steel, such as yield strength, ultimate strength, and elastic modulus, are shown in Table 4.

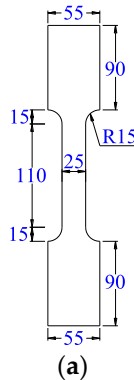
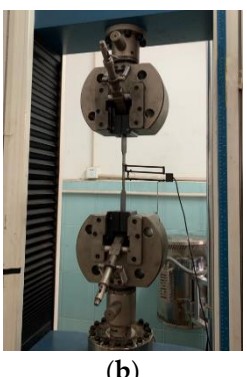

(**a**)                  (**b**)

**Figure 5.** Material test of steel. (**a**) Dimension (in mm). (**b**) Test machine.

**Table 4.** The properties of the steel coupons.

| Material | Steel Grade | $t$ (mm) | $f_y$ (MPa) | $f_u$ (MPa) | $E_s$ (MPa) |
|---|---|---|---|---|---|
| Steel tube | Q235 | 3.75 | 269 | 445 | 208,305 |
| Connection plate | Q235 | 3.75 | 258 | 402 | 186,208 |
| Stiffener | Q235 | 5.75 | 372 | 467 | 187,355 |

**Note:** $t$ is the thickness of the steel; $f_y$ is the yield strength of the steel; $f_u$ is the ultimate strength of the steel; $E_s$ is the elastic modulus of the steel.

## 2.3. Loading Device and Scheme

The loading device was a 500 T electro-hydraulic servo pressure test machine (Jilin guanteng Automation Technology Co., Ltd, Jilin, China). Both ends of the L-shaped column were hinged. The spherical hinge device was used to release the restraint, as presented in Figure 6.

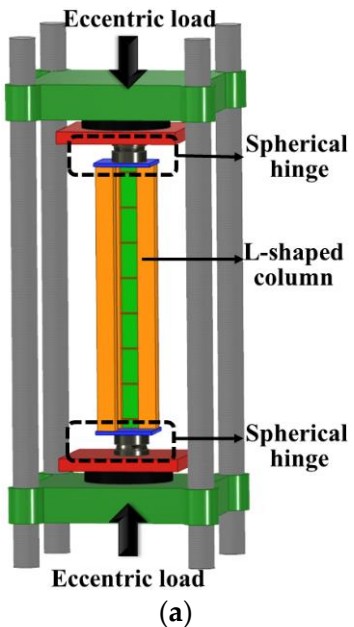
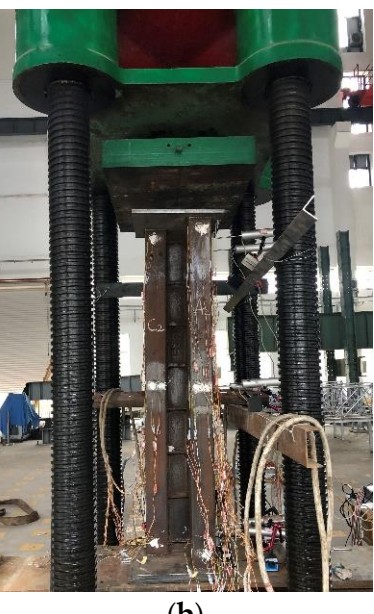

(**a**)                  (**b**)

**Figure 6.** Loading device. (**a**) 3D model. (**b**) On-site photo.

To identify the loading point of the specimen, it was necessary to obtain the cross-sectional centroid of the L-shaped column. In this paper, the geometric cross-section method was used to calculate the centroid of L-shaped columns based on the T/CECS825 [37]. As shown in Figure 7a, the corner point of column B is the coordinate origin O. The direction along the edge of the column limb is defined as two perpendicular coordinate axes in the plane. The coordinates of the cross-sectional centroid can be obtained by calculating the

static distances $S_x$ and $S_y$ of the cross-section to the $x$-axes and $y$-axes. The formula for calculating the coordinates of the centroid is presented in (1)–(2). To apply the bidirectional bending moment in the $X$-axis and $Y$-axis directions, the loading point of the specimen is 80 mm away from the centroid along the $X'$ axis, see Figure 7b.

$$x_c = \frac{S_y}{A} = \frac{\int_A x \, dA}{A} \tag{1}$$

$$y_c = \frac{S_x}{A} = \frac{\int_A y \, dA}{A} \tag{2}$$

where $x_c$, $y_c$ is the coordinates of the centroid from the $y$-axis and $x$-axis, respectively. $S_x$, $S_y$ is the static distance of the cross-section to the $x$-axis and $y$-axis, respectively. $A$ is the cross-sectional area enclosed by the geometric outer contour of the steel tubes and connection plates. $dA$ is the micro area taken at the coordinates $(x, y)$ within the cross-section.

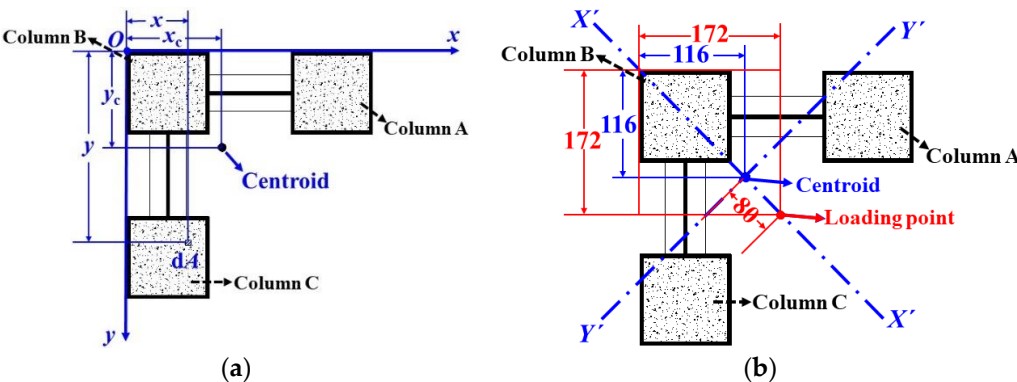

(**a**)  (**b**)

**Figure 7.** Position of the centroid and loading point. (**a**) Calculation of the centroid. (**b**) Loading point (in mm).

The loading scheme contains pre-loading and formal loading. During pre-loading, the load was applied at a rate of 20 kN/min to 10% of the yield load estimated by the numerical model. The operating status of the measurement equipment needs to be checked. Besides, the main purpose of pre-loading is to confirm that the spherical hinge coincides with the loading point of the specimen. The formal loading contains three stages. During the elastic stage, the load is gradually applied at intervals of 200 kN. During the yield stage, the load is applied at intervals of 100 kN. During the failure stage, the force control is replaced using displacement control with a loading rate of 0.3 mm/min. The test stopped when the load dropped to 85% of the peak load.

*2.4. Measurement Arrangement*

The linear variable displacement transducers (LVDTs) (Tjweekend, Tianjin, China) were used to monitor the displacement variation of the specimens, as shown in Figure 8. Twelve lateral LVDTs (H1–H12) were used to monitor the bending deformation of the specimen at both ends and in the middle of the column. Four vertical LVDTs (V1–V4) were arranged to monitor the compressive deformation of the specimen. To monitor the yielding and micro-deformation of the steel tubes, lateral and vertical strain gauges were arranged at the measurement points of the three mono columns and the connecting plates. The measurement points are divided into three zones along with the height of the column, see Figure 8b. Figure 8c provided details of the measurement points of the strain gauges and the orientation numbers of the columns.

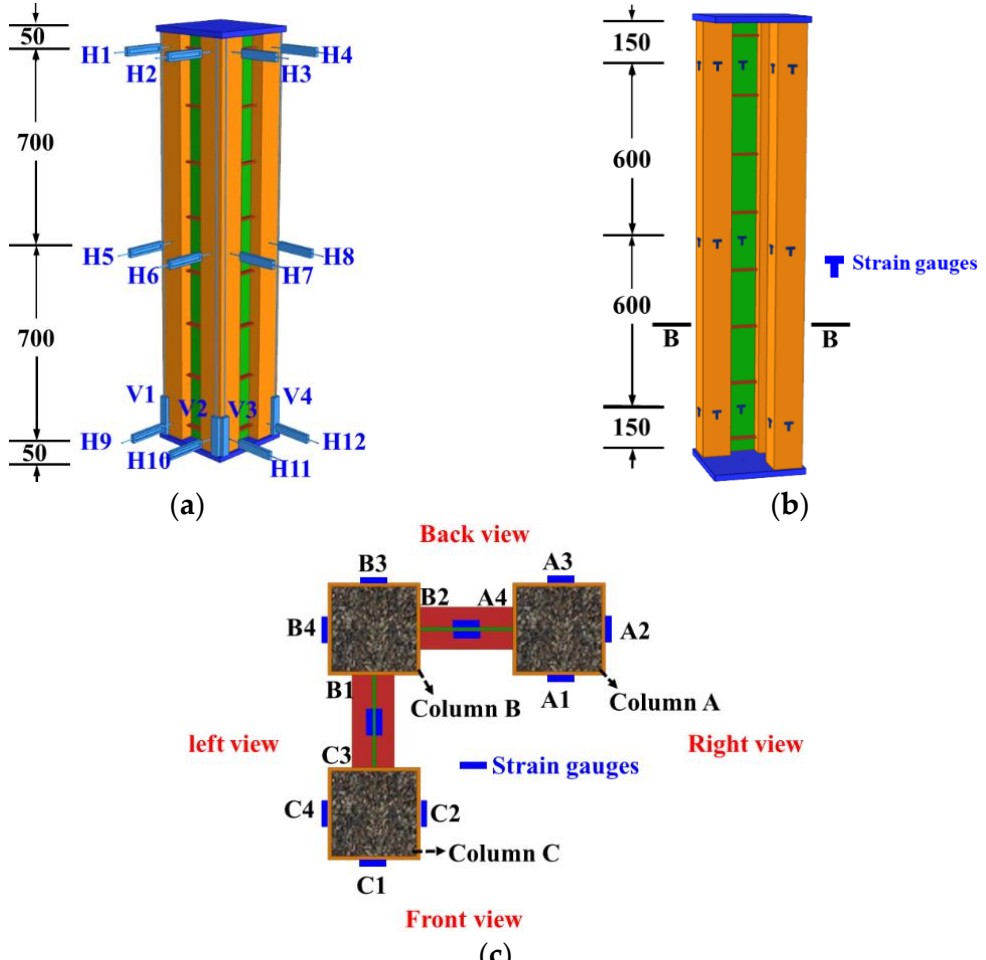

**Figure 8.** The layout of the measurement. (**a**) LVDT. (**b**) Strain gauges. (**c**) B-B cross-section.

## 3. Experimental Phenomenon

The coordinates and orientation numbers of the column cross-section are marked in Figures 7b and 8c to facilitate the presentation of the experimental phenomenon. According to the deformation process of the L-shaped column under eccentric compression, the failure mode can be divided into three stages: elastic stage, elastoplastic stage, and failure stage. There was no significant experimental phenomenon of the specimens in the elastic stage due to the micro elastic deformation of steel and RAC.

The specimen entered the elastoplastic stage as the load increased. The mono columns of specimens P80-0~P80-100 were slightly bent around the $Y'$ axis due to the bending moment. The specimen deformed slowly as the load gradually increased to the peak load. The local buckling of column A and column C began to occur at the middle or upper end of the L-shaped column, as presented in Figure 9a,b. There was basically no local buckling of the steel tube in column B. This is because the section of column B was subjected to the tensile force under the eccentric load. Besides, the connection plates on both sides formed the confinement on column B. For specimen P80-K, the hollow steel tube began to exhibit inward buckling due to the absence of RAC inside, as shown in Figure 9c.

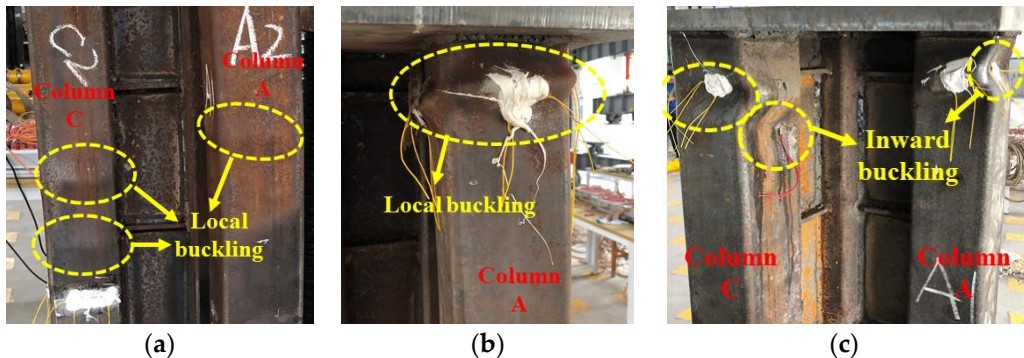

**Figure 9.** Typical phenomenon in elastoplastic stage. (**a**) Middle of the column. (**b**) Upper end of the column. (**c**) Inward buckling of steel tube.

In the failure stage, the specimen bent and deformed rapidly in a short time due to the eccentric load. Two compression failure modes appeared in the specimens P80-0~P80-100. In one of the failure modes, the failure section appeared in the middle of the column, including specimens P80-0, P80-20, and P80-100. The column section was bent around the *Y'* axis. Columns A and C presented multiple local bucklings in the middle of the column, as shown in Figure 10a,e,f. There was no obvious buckling deformation on the connection plate, indicating that the mono columns could be cooperatively deformed. In another failure mode, the failure section occurred at the upper end of the column. The obvious local buckling of columns A and C was concentrated at the upper end, as shown in Figure 10b–d. For specimen P80-K, the inward buckling occurred at the upper end of columns A and C. According to the comparison of the failure mode, the deflection of the specimen with the failure section at the column end was less than that of the specimen with the failure section in the middle of the column. Besides, the comparison of the failure mode between the L-shaped columns filled with RAC and without RAC indicated that the core concrete could effectively inhibit the inward buckling of steel tubes.

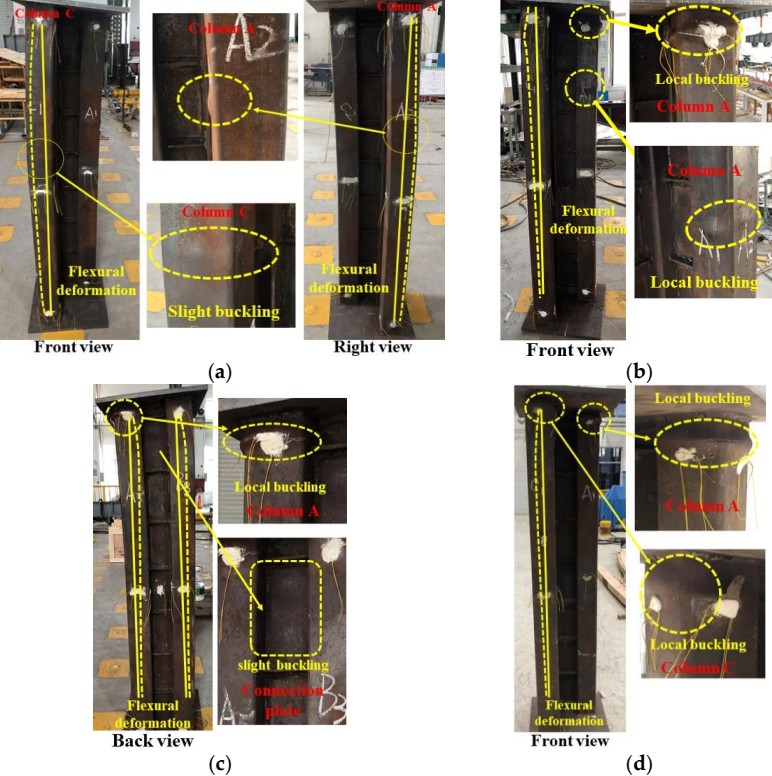

**Figure 10.** *Cont*.

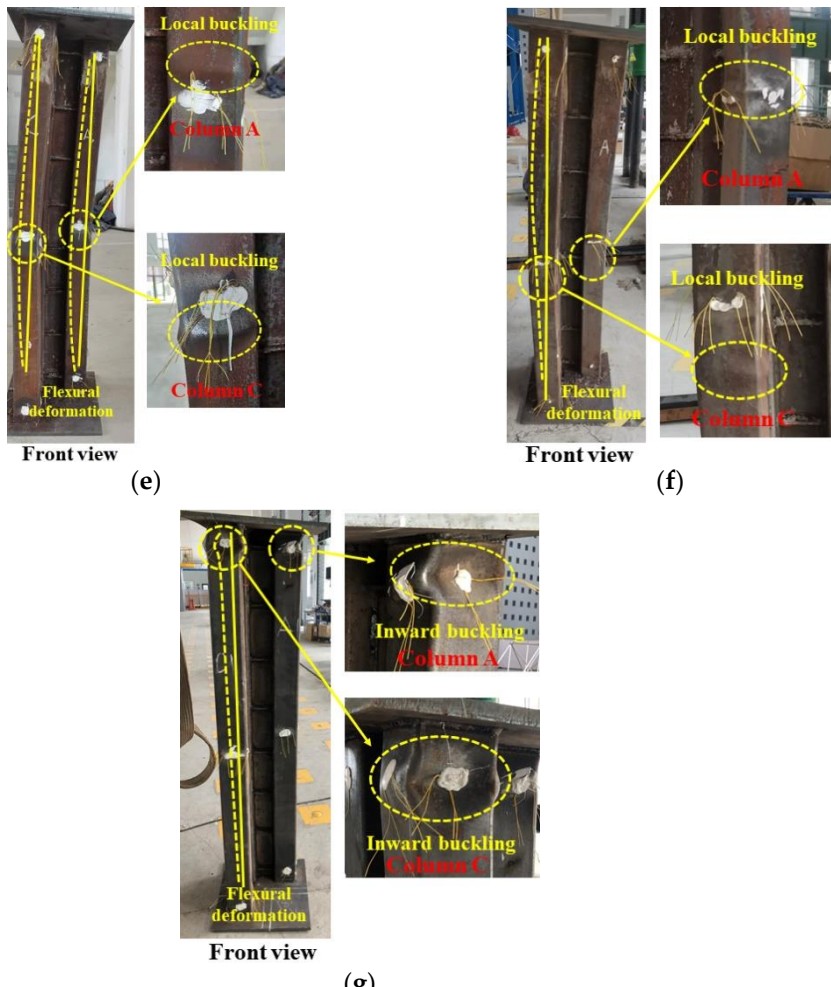

**Figure 10.** Failure phenomenon of the specimens. (**a**) P80-0. (**b**) P80-20. (**c**) P80-40. (**d**) P80-60. (**e**) P80-80. (**f**) P80-100. (**g**) P80-K.

All the L-shaped columns filled with RAC exhibited integral flexural deformation. The failure mode of the specimens presented the compression-bending damage of columns A and C. The effect of different replacement ratios on the failure mode was not significant. The L-shaped column without RAC was subject to premature loss of load capacity due to inward buckling of the steel tubes.

## 4. Test Results and Analysis

To facilitate the comparison of the mechanical properties of the specimens at different stages, the loading-displacement curves were divided into the elastic stage, elastoplastic stage, and failure stage. The critical point between the elastic stage and the elastoplastic stage is the yield point, which can be obtained by the "farthest point method" [38]. The critical point between the elastoplastic stage and the failure stage is the peak point, which corresponded to the peak load. The point corresponding to 85% of the peak load in the failure is the ultimate point. Besides, the ductility index (*DI*) was defined as Equation (3) to characterize the ductility of the L-shaped column. The secant stiffness ($S_y$) of the yield point was utilized to characterize the stiffness of the L-shaped column, which was defined as in Equation (4).

$$DI = \frac{\Delta_u}{\Delta_y} \tag{3}$$

$$S_y = \frac{N_y}{\Delta_y} \tag{4}$$

where $\Delta_u$ is the ultimate displacement, $\Delta_y$ is the yield displacement, $N_y$ is the yield load of the specimen.

The data of the yield point, peak point, and ultimate point are shown in Table 5. All the loading-displacement curves of specimens are presented in Figure 11. The comparison of the loading-displacement curves indicated that specimen P80-0 had the highest stiffness, and the stiffness of the L-shaped columns decreased significantly as the replacement ratio of RAC increased. Compared to the yield point, secant stiffness Sy of specimen P80-0, the specimens P80-20, P80-40, P80-60, P80-80, P80-100, and P80-K decreased by 18.4%, 43.4%, 51.2%, 57.1%, 68.5%, and 72.7%, respectively. The above data indicated that the stiffness of the specimens decreased linearly with the RCA replacement ratio of the in-filled RAC increased from 0 to 40%. The stiffness of the specimens decreased slowly as the replacement ratio of RCA increased from 40% to 100%. By comparing P80-K with other specimens, the filling of RAC in the steel tube effectively improved the stiffness of the L-shaped column.

**Table 5.** The characteristic points of the specimens.

| Specimens | Yield Point | | Peak Point | | Ultimate Point | | DI | $S_y$ (kN/mm) |
|---|---|---|---|---|---|---|---|---|
| | $\Delta_y$ (mm) | $N_y$ (kN) | $\Delta_p$ (mm) | $N_p$ (kN) | $\Delta_u$ (mm) | $N_u$ (kN) | | |
| P80-0 | 1.46 | 1450 | 5.75 | 1904 | 16.68 | 1618 | 11.42 | 993 |
| P80-20 | 1.54 | 1248 | 6.28 | 1820 | 13.42 | 1547 | 8.71 | 810 |
| P80-40 | 1.93 | 1085 | 6.80 | 1652 | 10.98 | 1404 | 5.69 | 562 |
| P80-60 | 2.14 | 1038 | 7.16 | 1580 | 11.54 | 1343 | 5.39 | 485 |
| P80-80 | 2.31 | 985 | 7.59 | 1510 | 12.22 | 1284 | 5.29 | 426 |
| P80-100 | 2.90 | 909 | 8.36 | 1449 | 13.45 | 1232 | 4.64 | 313 |
| P80-K | 2.63 | 713 | 7.92 | 1222 | 13.14 | 1039 | 5.01 | 271 |

**Note:** $\Delta_y$ is the yield displacement, $N_y$ is the yield load, $\Delta_p$ is the peak displacement, $N_p$ is the peak load, $\Delta_u$ is the ultimate displacement, $N_u$ is the ultimate load, *DI* is the ductility index, $S_y$ is the secant stiffness of the yield point.

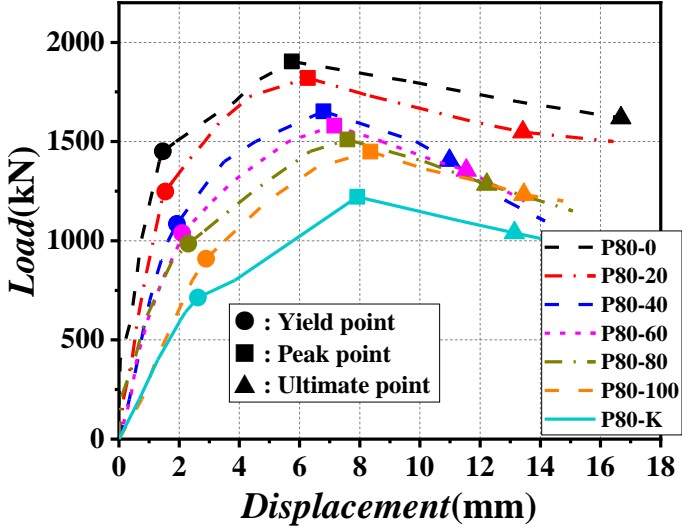

**Figure 11.** Axial load-displacement curve.

Besides, the variation of bearing capacity with the replacement ratio is displayed in Figure 12. The bearing capacity gradually decreased with the increase of the RCA replacement. When the RCA replacement ratio increased from 0% to 40%, the yield load and peak load decreased by 25.2% and 13.2%, respectively. When the RCA replacement ratio increased from 40% to 100%, the yield load and peak load decreased by 16.2% and 12.3%, respectively. This indicated that the bearing capacity decreased more rapidly when the RCA replacement ratio was below 40%. When the RCA replacement ratio exceeded 40%, the bearing capacity decreased at a slower rate. Besides, the yield load and peak load of

specimen P80-K were 21.6–50.8% and 15.7–35.8% lower than other specimens, respectively. This indicated that the filling of concrete can increase the bearing capacity by preventing the steel tube from buckling inward and contributing to the vertical compressive strength.

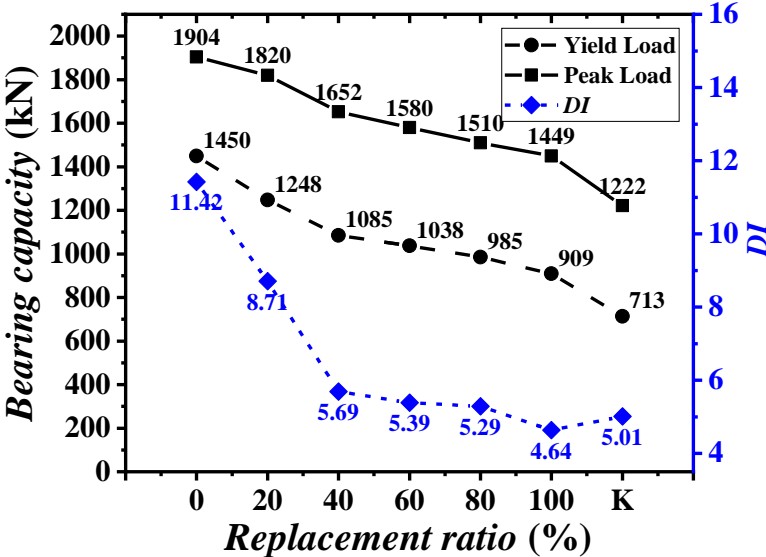

**Figure 12.** Bearing capacity and ductility.

From the variation curve of *DI* in Figure 12, the ductility of the L-shaped column decreased significantly with the increase of the RCA replacement ratio. When the RCA replacement ratio increased from 0% to 40%, the DI decreased by 50.2%. When the RCA replacement ratio increased from 40% to 100%, the DI decreased by 18.5%. The above data indicated that the ductility of the specimens decreased significantly when ordinary concrete was replaced by RAC. This phenomenon was especially obvious when the replacement ratio of RCA was less than 40%. When the replacement ratio of RCA exceeded 40%, the effect of the elevated replacement ratio on the ductility of the specimens decreased significantly. Besides, The *DI* of specimen P80-K was close to that of specimen P80-80, which indicated that the ductility of specimens with a higher RAC replacement ratio was closer to that of the L-shaped column composed of hollow steel tubes. Therefore, RAC with less than 40% RCA replacement ratio is recommended to be applied in L-shaped columns.

## 5. Finite Element Analysis

### 5.1. Finite Element Models

In this paper, the finite element analysis software ANSYS was used to simulate the test process. The peak load and failure mode of the RACFST obtained by finite element simulation were compared with the test results to verify the effectiveness of the numerical simulation method. The steel, connecting plate, and stiffener were modeled by SHELL181 elements, and the RAC was simulated by SOLID65 elements. Surface–surface contact between the steel tube and the inner concrete was used to simulate the interaction between steel and concrete. The concrete side as the "target surface" was simulated by the TARGE170 element, and the steel tube side as the "contact surface" was simulated by the CONTA173 element. Constraining the *X*, *Y*, and *Z* degrees of freedom of all nodes at the bottom of the RACFST and all nodes at the top of the RACFST was coupled with the load application position and constrained *X* and *Y* degrees of freedom. This is consistent with the experimental boundary conditions. The modeling method referred to previous literature [20]. The finite element model is presented in Figure 13.

In the modeling, the constraint effect of steel tubes on concrete was considered [39]. For the constitutive relationship of recycled concrete, the reader is referred to [40].

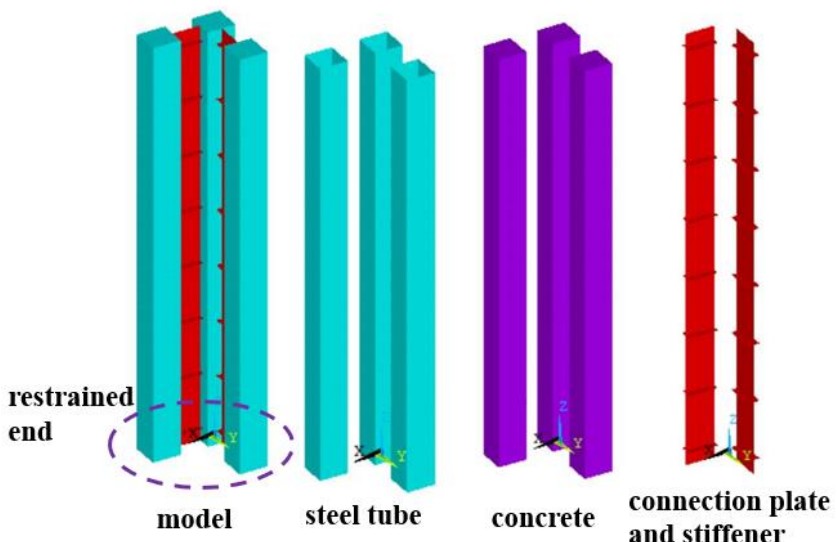

**Figure 13.** Finite element models.

## 5.2. The Comparison of the Axial Load-Displacement Curve

The load-vertical displacement comparison curve obtained from numerical simulation and test results is shown in Figures 14 and 15. It can be seen that the stiffness of the two curves agreed well. The comparison bearing capacity of the yield load and peak load of RACFST are listed in Table 6. For RACFST with different RCA replacement ratios, the average yield load from numerical simulation and test results was 1.10, and the average peak load was 1.05. It can be seen that the bearing capacity by numerical simulation was in good agreement with the experimental results. In conclusion, the numerical simulation method used in this paper can effectively simulate the eccentric bearing capacity of the RACFST.

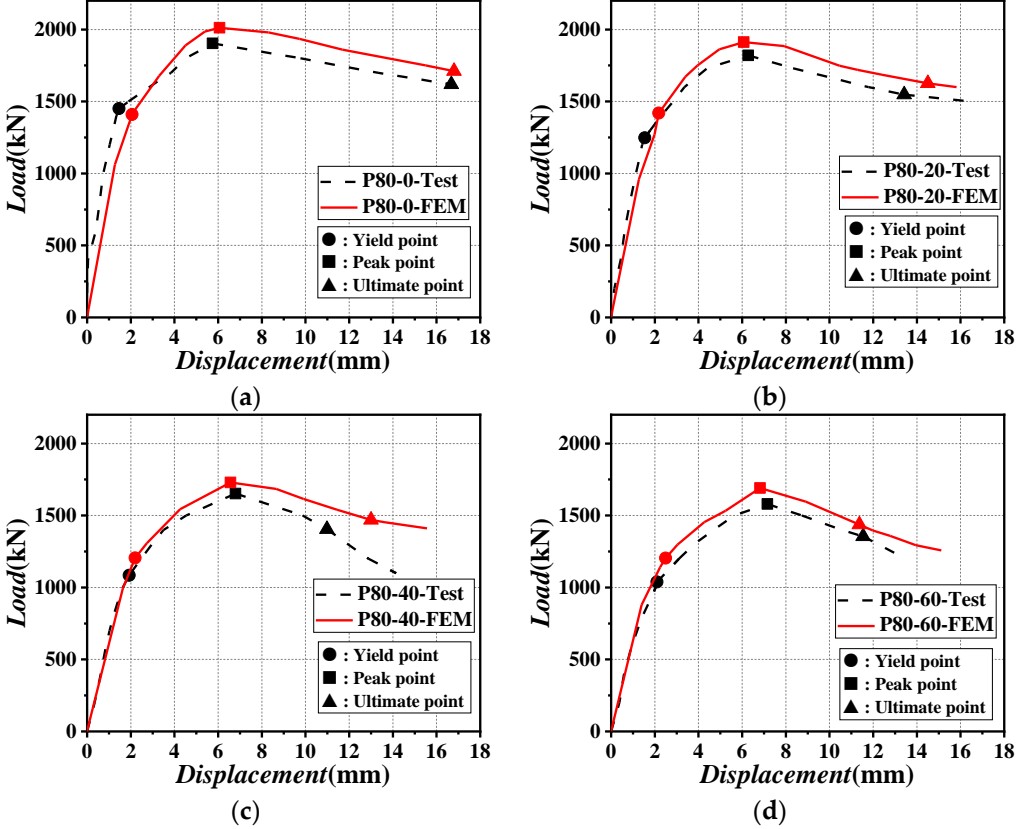

**Figure 14.** *Cont.*

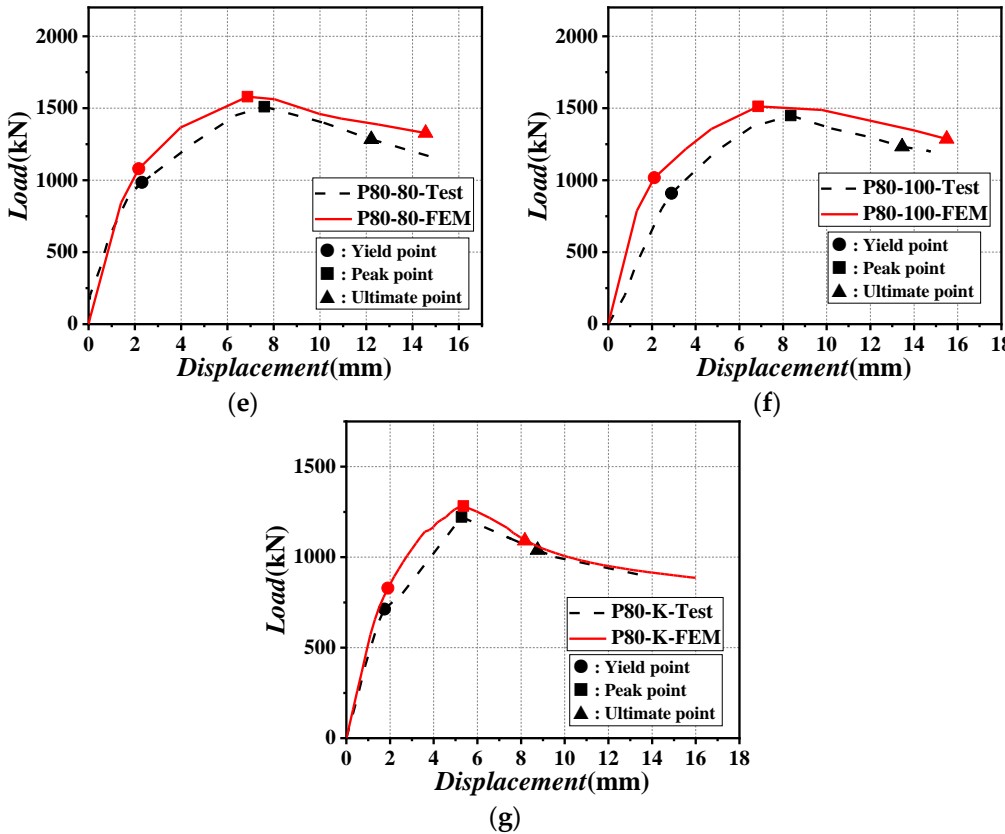

**Figure 14.** Comparison of the axial load-displacement curve. (**a**) P80-0. (**b**) P80-20. (**c**) P80-40. (**d**) P80-60. (**e**) P80-80. (**f**) P80-100. (**g**) P80-K.

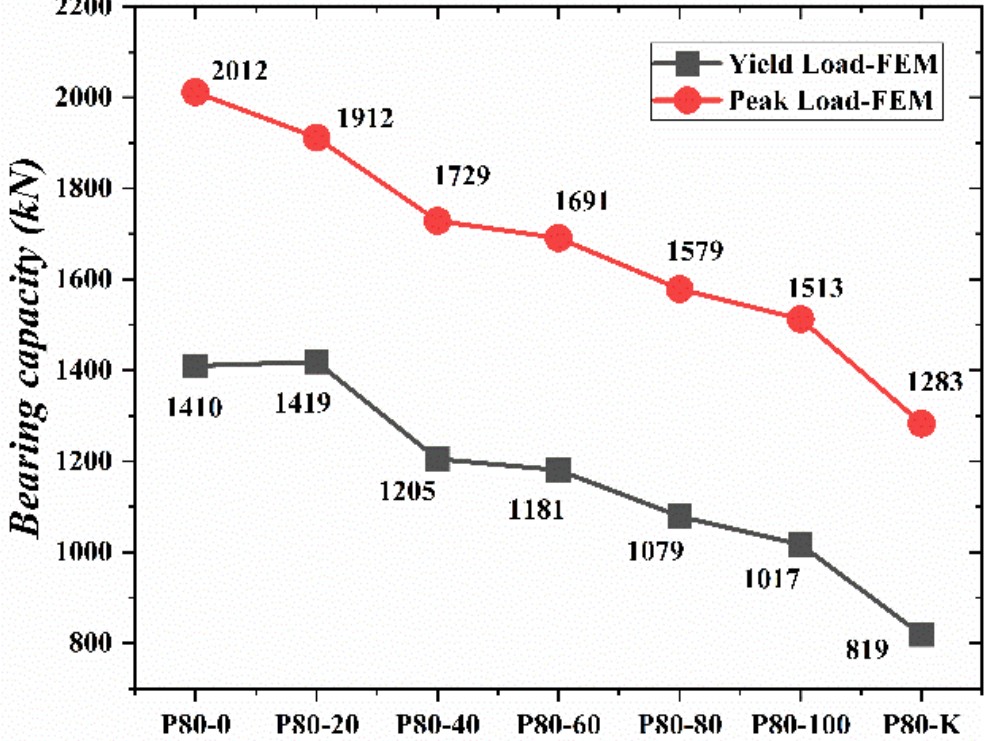

**Figure 15.** Bearing capacity of FEM.

**Table 6.** Comparison of the bearing capacity.

| Specimens | Yield Point | | $N_{y\text{-}FEM}/N_{y\text{-}test}$ | Peak Point | | $N_{p\text{-}FEM}/N_{p\text{-}test}$ |
|---|---|---|---|---|---|---|
| | $N_{y\text{-}FEM}$ (kN) | $N_{y\text{-}test}$ (kN) | | $N_{p\text{-}FEM}$ (kN) | $N_{p\text{-}test}$ (kN) | |
| P80-0 | 1410 | 1450 | 0.97 | 2012 | 1904 | 1.06 |
| P80-20 | 1419 | 1248 | 1.14 | 1912 | 1820 | 1.05 |
| P80-40 | 1205 | 1085 | 1.11 | 1729 | 1652 | 1.05 |
| P80-60 | 1181 | 1038 | 1.14 | 1691 | 1580 | 1.07 |
| P80-80 | 1079 | 985 | 1.10 | 1579 | 1510 | 1.05 |
| P80-100 | 1017 | 909 | 1.11 | 1513 | 1449 | 1.04 |
| P80-K | 819 | 713 | 1.14 | 1283 | 1222 | 1.05 |
| | *MV* | | 1.10 | *MV* | | 1.05 |

**Note**: $N_{y\text{-}FEM}$ is the yield load of FEM, $N_{y\text{-}test}$ is the yield load of test results, $N_{p\text{-}FEM}$ is the peak load of FEM, $N_{p\text{-}test}$ is the peak load of test results, *MV* is the mean value.

### 5.3. The Verification of the Failure Mode

The comparison failure modes of each specimen are shown in Figure 16. The overall deformation of each specimen basically shows the bending deformation around the weak axis. The failure modes of each specimen in FEM and test were similar. The local buckling position of each specimen in FEM and test were similar. There was a slight difference between the FEM results and the test results, which may be caused by the failure to control the speed of applying the load during the test. It may also be caused by the fabrication deviation of the specimen or the non-uniformity of the filling concrete. In general, the failure modes of the FEM results were generally similar to the test. The above modeling method can provide a reference for exploring L-shaped columns composed of RACFST.

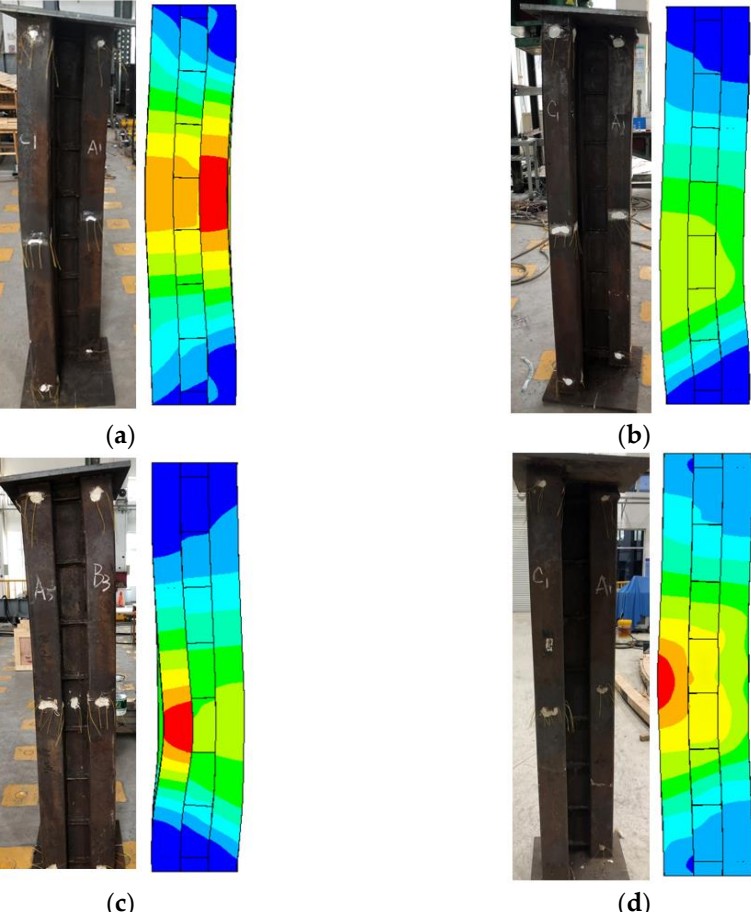

(a)  (b)  (c)  (d)

**Figure 16.** *Cont.*

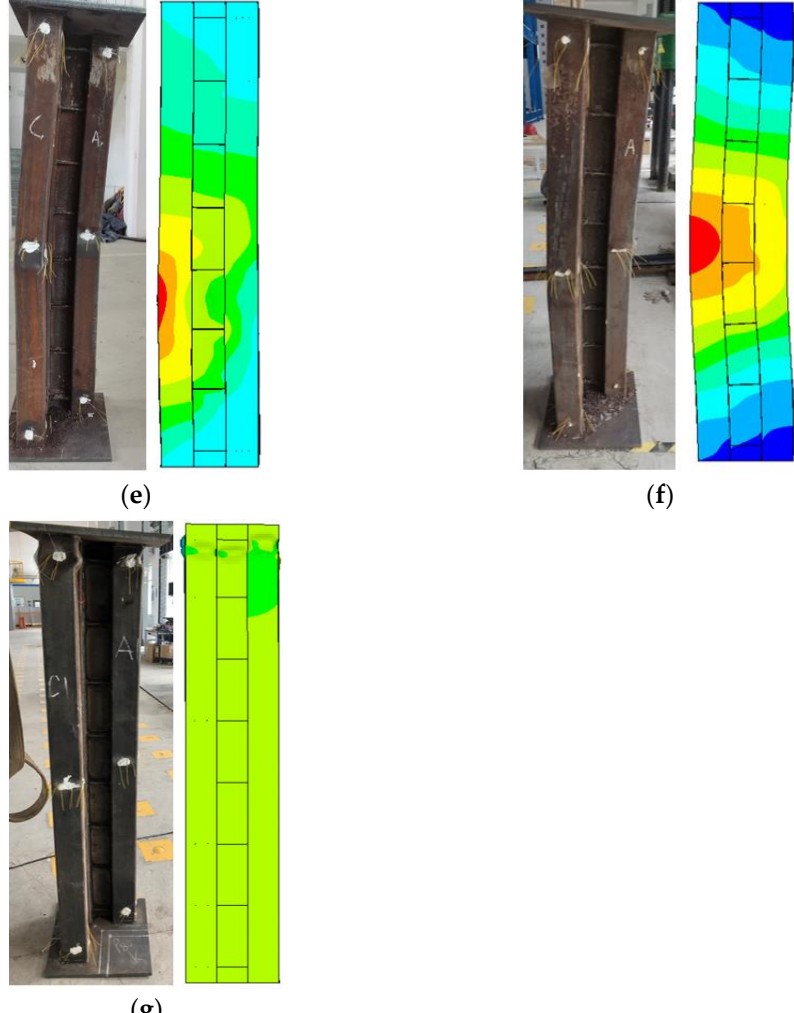

**Figure 16.** Comparison of the failure mode. (**a**) P80-0. (**b**) P80-20. (**c**) P80-40. (**d**) P80-60. (**e**) P80-80. (**f**) P80-100. (**g**) P80-K.

## 6. Parametric Analysis

### 6.1. The Effect of the Eccentricity

In this paper, five different eccentric loading positions were selected to study the eccentric mechanical properties of RACFST. The eccentric positions were all on the OX'axis, as shown in Figure 7. The eccentricities of loading points were 0 mm (centroid), 40 mm, 80 mm, 120 mm, and 160 mm. The load-eccentricity curve of RACFST is shown in Figure 17. It can be seen from Figure 17 that the peak load of RACFST decreased with the increase of eccentricity, and the greater the RAC ratio, the greater the decrease in peak load. When the replacement ratio of RAC was 0% and the loading position increased from 0 mm to 160 mm, the peak load of RACFST decreased by about 35%. When the replacement ratio of RAC was 100%, the peak load of RACFST decreased by about 45%.

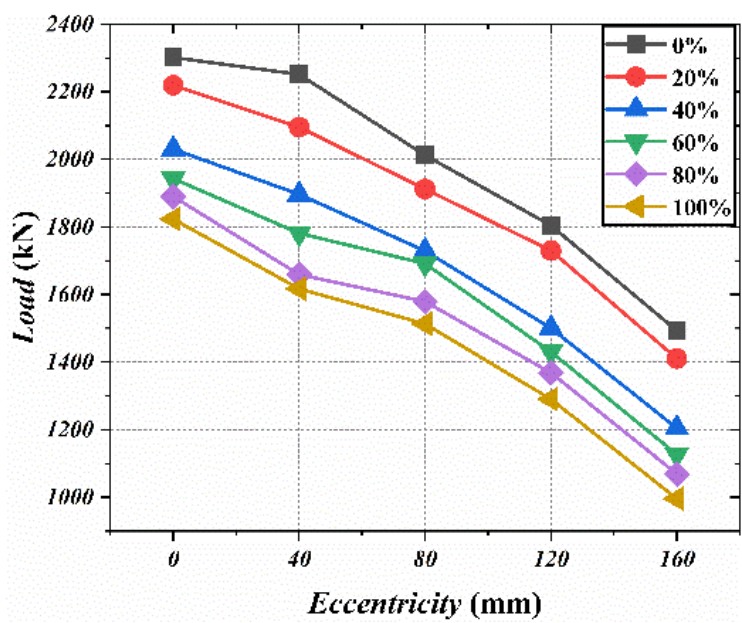

**Figure 17.** Load-eccentricity curve.

### 6.2. The Effect of the Steel Strength

In this paper, Q235, Q345, Q390, Q420, and Q460 were selected as variables to study the effect of steel strength on the eccentric mechanical properties of RACFST. The load-steel products curve of RACFST is shown in Figure 18. It can be seen from Figure 18 that with the improvement of steel strength, the peak load of RACFST decreased. When the steel strength was increased by one level, the amplitude of the peak load of RACFST increased; it could be increased by about 35%. Thus, the peak load of RACFST can be improved by improving the steel strength.

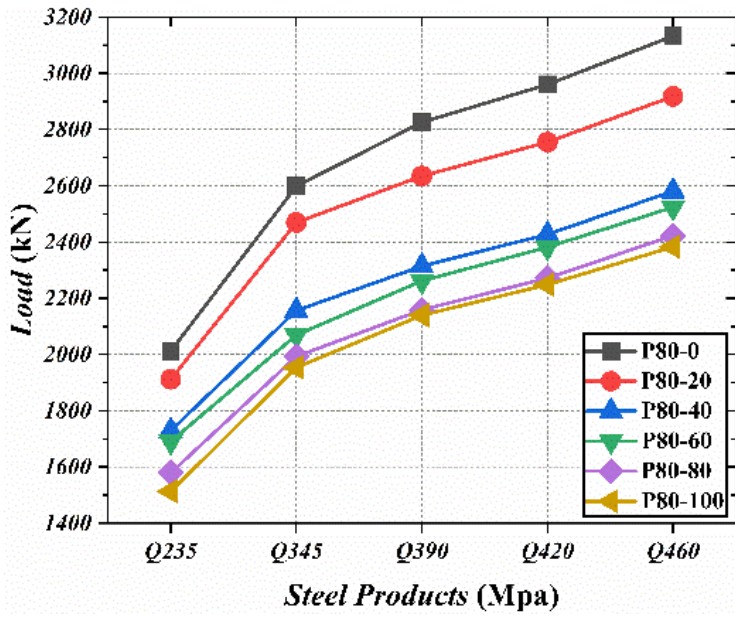

**Figure 18.** Load-steel products curve.

### 6.3. The Effect of the Thickness of the Steel

In order to study the influence of steel thickness on the eccentric mechanical properties of RACFST, nine different steel thicknesses of 3.75 mm, 5.75 mm, 7.75 mm, 9.75 mm, 11.75 mm, 13.75 mm, 15.75 mm, 17.75 mm, and 19.75 mm were selected to simulate the peak

load. The steel thickness mentioned here refers to the thickness of the steel tube connecting the plate and stiffener. The load-steel thickness curve of RACFST is shown in Figure 19. Figure 19 shows that with the increase in steel thickness, the peak load of RACFST shows an upward trend. When the steel thickness increased from 3.75 mm to 19.75 mm, the increased amplitude of the peak load of RACFST under different RAC replacement ratios was 220%~317%, and the peak load increased with the RAC replacement ratio. This shows that the peak load of RACFST can be improved by increasing the steel thickness.

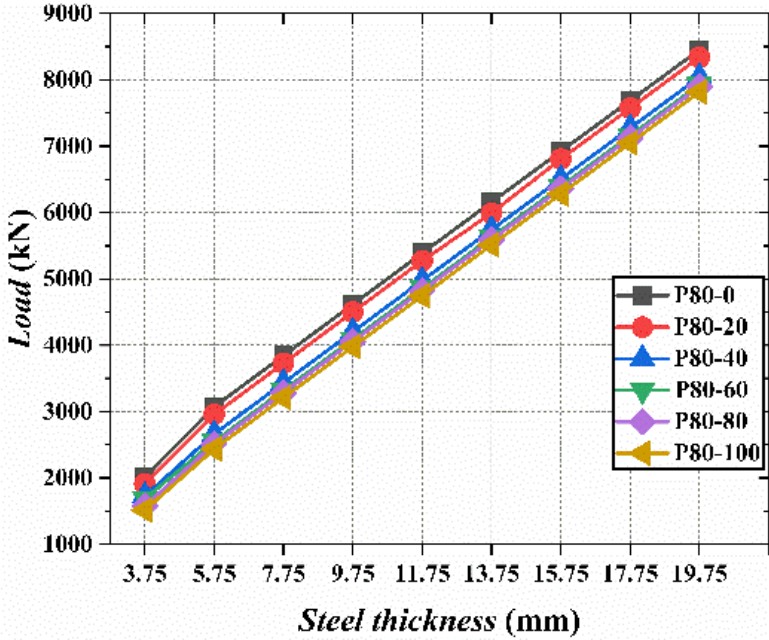

**Figure 19.** Load-steel thickness curve.

## 7. Conclusions

In this paper, the eccentric compressive performance of the L-shaped column composed of recycled aggregate concrete-filled steel tubes was investigated. The eccentric compression performance of RACFST with the different replacement ratios of recycled coarse aggregate was explored and compared with the hollow L-shaped column. In addition, a numerical simulation of eccentricity, steel strength, and steel thickness was conducted. The research conclusions are summarized as follows:

(1) The peak load of the hollow L-shaped column is lower than that of the L-shaped column composed of recycled aggregate concrete-filled steel tubes with different coarse aggregate replacement ratios, which shows that the recycled concrete improves the peak load of the RACFST. Compared with the hollow L-shaped column, the peak load of the RACFST with a 0% replacement ratio of RCA increased by 35.82%, and the peak load of the RACFST with a 100% replacement ratio of RCA increased by 15.67%.

(2) Under eccentric loading, with an increased RCA replacement ratio, the stiffness of the RACFST decreases. When the replacement ratio of RCA exceeds 40%, the reduction rate of stiffness slows down.

(3) Under eccentric loading, with an increased RCA replacement ratio, the ductility of the RACFST decreases. When the replacement ratio of RCA exceeded 40%, the effect of the elevated replacement ratio on the ductility of the specimens decreased significantly.

(4) With the increase of eccentricity, the bearing capacity of the RACFST decreases gradually.

(5) With the improvement of steel strength, the bearing capacity of the RACFST shows an upward trend; that is, the bearing capacity of the RACFST can be improved by strengthening the steel strength.

(6) With the increase in steel thickness, the bearing capacity of the RACFST shows an upward trend; that is, the bearing capacity of the RACFST can be improved by increasing the steel thickness.

**Author Contributions:** T.M.: conceptualization, testing, software, writing—review and editing; Z.C.: supervision, visualization; Y.D.: validation, formal analysis, methodology; T.Z.: investigation, resources; Y.Z.: data curation, writing—original draft and review. All authors have read and agreed to the published version of the manuscript.

**Funding:** This work is sponsored by The National Key R&D Program of China (Chen, Z: 2019YFD1101005), the Science and Technology Project of the Hebei Education Department (Ma, T: Grant No.QN2021003), and the Science and Technology Project of the North China Institute of Aerospace Engineering (Ma, T: ZD-2022-05).

**Institutional Review Board Statement:** Not applicable.

**Informed Consent Statement:** Not applicable.

**Data Availability Statement:** Not applicable.

**Acknowledgments:** The authors of the paper appreciate the support from the National Natural Science Foundation of China (Grant No. 51808182) and the China Postdoctoral Science Foundation (Grant No. 2020M670680).

**Conflicts of Interest:** The authors declare that they have no known competing financial interests or personal relationships that could have appeared to influence the work reported in this paper.

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
