# Peer review of "Mechanical Properties of L-Shaped Column Composed of RAC-Filled Steel Tubes under Eccentric Compression"

_metals, doi:10.3390/met12060953_

Round 1

Reviewer 1 Report

The following are the comments that the authors should address. 

  1. In the Introduction, please discuss on the coda requirements of using RAC for such applications. 
  2. Information on how the RAC concrete mix was designed is not provided. Please include this. 
  3. There is a significant drop when the RAC content was increased from 20% to 40%. What may have caused this? Please discuss.
  4. Please present the properties of the aggregates (fine and coarse aggregates) including the RAC. 
  5. How were the constitutive models for the RAC and the virgin aggregates varied in the FEM? How were the parameters inputted?
  6. The quality of figures, especially the graphs should be improved.  

Author Response

Reviewer #1:

Thank you for providing the reviews for our paper: metals-1724042. We would like to thank the reviewers for their time and consideration. Herein we explain how we revised the paper based on those comments and recommendations. Besides, we have revised the manuscript (in red) in the Revised Manuscript file. Thank you for the opportunity to revise the manuscript.

Comment 1: In the Introduction, please discuss on the coda requirements of using RAC for such applications. 

Response 1: Thanks for your comment. In the introduction, it is discussed that the initial defects of recycled concrete filled steel tube caused by recycled concrete strength and pouring compactness should be minimized. At present, the strength limit of recycled concrete filled steel tube is not clearly stated. Therefore, the strength limit of recycled concrete filled steel tube is also one of the follow-up research contents of this paper.

Comment 2: Information on how the RAC concrete mix was designed is not provided. Please include this.

Response 2: Thanks for your comment. The mix proportion is based on the mix proportion of ordinary concrete C30 to explore the influence of recycled aggregate replacement ratio. Therefore, the mix proportion design in Table 2 is based on the mix proportion design of current commercial C30 concrete.

Comment 3: There is a significant drop when the RAC content was increased from 20% to 40%. What may have caused this? Please discuss.

Response 3: Thanks for your advice. Due to the test site, the curing age of RAC is different. The curing age of RAC test block with RCA replacement ratio of 40%-100% is 28 days. The curing age of RAC test block with RCA replacement ratio of 0%-20% is 55 days. Curing age is listed in Table 3

Comment 4: Please present the properties of the aggregates (fine and coarse aggregates) including the RAC.

Response 4: Thanks for your advice. In section 2.2.1, we had corrected the problem as you suggested.

Comment 5: How were the constitutive models for the RAC and the virgin aggregates varied in the FEM? How were the parameters inputted?

Response 5: Thanks for your comment. In the modeling, the constraint effect of steel tube on concrete is considered in this paper [39]. The constitutive relationship of recycled concrete is referred to refenerce [40].

[39] Y Yang, L Han. Compressive and flexural behaviour of recycled aggregate concrete filled steel tubes (RACFST) under short-term loadings[J].Steel & Composite Structures,2006, 6 (3): 257-284.

[40] Q Li,Z Chen,Y Du,et al.Study on constitutive model of core concrete of recycled aggregate concrete filled steel tubular columns under compression,Industrial Construction, 2021,51(5):108-115,15.

Comment 6: The quality of figures, especially the graphs should be improved.

Response 6: Thanks for your comment. We have improved and modified the pictures in this paper.

Reviewer 2 Report

The paper present a research about a specific way to realize welded built-up composite columns composed of several CFTs; the infill is made of recycled concrete aggregates or RCA. The introduction and presentation and conclusions are good. There are however some problems with formulation in English and with editorial details; all these are gathered hereunder in detailed remarks.

One more fundamental remark, though it does not impair the quality of the work which is presented is that there is no explanation at all why the strength in compression of RCA is smaller than the one of concrete and why this strength decreases with increasing RCA replacement ratio. There may be several reasons and this should be explained, even briefly. And it should be mentioned that this effect may vary depending on the type of RCA, so that the results obtained are qualitative, with correct trends, but that there can be variations if other types of RCA are used.

Remarks of details

54: "An appropriate…" rather than "The appropriate…"

59 The special-shaped column composed of mono CFST column…rather columns    because it cannot be composed of a single column

63 and 65 same: columns    one can only link several things together, not a single one. In fact the proposed sections are welded built up sections composed of several CFST. It might be better to say so.

88 not "occurs" but "undergoes"

104 "To study the mechanical properties of shaped columns subjected to the combined action of axial pressure and bending moments". There is no verb in the sentence. Meaning?

112 "…element model (FEM) was verified by…" rather: " …element model (FEM) results were compared with…"

In Figure 3: "stiffener interval" not "stiffer interval"

118 " The mono column ie the RAC-filled steel tube" sentence without verb. Meaning?

120 " The L-shaped column is welded by three mono column". No, rather: " The L-shaped column is composed of three mono column connected together by means of two connection plates".

153     2.2.2 steel  rather Steel

Table 2             Es for steel plates is much lower than 205000. Why?

273   The numbers of the equations are (1) and (2), which have already been used before. Number (3) and (4) and make corrections in text.

277   data…are shown…"data" is plural

317 It is not Figure 1 but 11. And further down, 2 is 12, etc…Please correct all and in text accordingly

Figure 19 cannot be read, because of bad quality of notations. Please correct.

Author Response

Reviewer #2:

Thank you for providing the reviews for our paper: metals-1724042. We would like to thank the reviewers for their time and consideration. Herein we explain how we revised the paper based on those comments and recommendations. Besides, we have revised the manuscript (in red) in the Revised Manuscript file. Thank you for the opportunity to revise the manuscript.

Comment 1: There is no explanation at all why the strength in compression of RCA is smaller than the one of concrete and why this strength decreases with increasing RCA replacement ratio.

Response 1: Thanks for your comment. In section I "Introduction", the factors affecting the strength reduction of recycled concrete are explained, mainly referring to reference [28].

[28] Kim J . Influence of quality of recycled aggregates on the mechanical properties of recycled aggregate concretes: An overview[J]. Construction and Building Materials, 2022, 328:127071-.

Comment 2: 54: "An appropriate…" rather than "The appropriate…"

Response 2: Thanks for your comment. After revision , the ‘The appropriate…’ were changed into ‘An appropriate…’.

Comment 3: 59 The special-shaped column composed of mono CFST column…rather columns    because it cannot be composed of a single column

Response 3: Thanks for your advice. After revision, the ‘column’ were changed into ‘columns’.63 and 65 were revised together.

Comment 4: 88 not "occurs" but "undergoes"

Response 4: Thanks for your advice. After revision, the ‘occurs’ were changed into ‘undergoes’

Comment 5: 104 "To study the mechanical properties of shaped columns subjected to the combined action of axial pressure and bending moments". There is no verb in the sentence. Meaning?

Response 5: Thanks for your advice. After revision,the sentence were changed into “To study the mechanical properties of shaped columns under the combined action of axial pressure and bending moments.”

Comment 6: 112 "…element model (FEM) was verified by…" rather: " …element model (FEM) results were compared with…"

Response 6: Thanks for your comment. After revision,the sentence were changed into“The finite element model (FEM) were compared with the experimental results”.

Comment 7: In Figure 3: "stiffener interval" not "stiffer interval"

Response 7: Thanks for your comment. In Fig. 3 the ‘stiffer interval’ were changed into ‘stiffener interval’.

Comment 8: 118 " The mono column ie the RAC-filled steel tube" sentence without verb. Meaning?

Response 8: Thanks for your advice. After revision, the sentence were changed into“The mono column is the RAC-filled steel tube.”

Comment 9: 120 " The L-shaped column is welded by three mono column". No, rather: " The L-shaped column is composed of three mono column connected together by means of two connection plates".

Response 9: Thanks for your advice. After revision, the sentence were changed into“The L-shaped column is composed of three mono column connected together by means of two connection plates.”

Comment 10: 153     2.2.2 steel  rather Steel

Response 10: Thanks for your advice. the ‘steel’ were changed into ‘Steel’.

Comment 11: Table 2             Es for steel plates is much lower than 205000. Why?

Response 11: Thanks for your advice. This is caused by test errors such as fixture slip.

Comment 12: 273   The numbers of the equations are (1) and (2), which have already been used before. Number (3) and (4) and make corrections in text.

Response 12: Thanks for your comment. Corrected here.

Comment 13: 277   data…are shown…"data" is plural

Response 13: Thanks for your comment. Corrected here.

Comment 14: 317 It is not Figure 1 but 11. And further down, 2 is 12, etc…Please correct all and in text accordingly

Response 14: Thanks for your comment. Corrected here.

Comment 15:Figure 19 cannot be read, because of bad quality of notations. Please correct.

Response 15: Thanks for your comment. Corrected here.